# GT-Space: Enhancing Heterogeneous Collaborative Perception with Ground Truth Feature Space

**Wentao Wang**[1]    **Haoran Xu**[1,2]    **Guang Tan**[1]*
[1] Sun Yat-sen University, Shenzhen Campus, Shenzhen, China
[2] Peng Cheng Laboratory (PCL), Shenzhen, China
{wangwt66, xuhr9}@mail2.sysu.edu.cn, tanguang@mail.sysu.edu.cn

## Abstract

In autonomous driving, multi-agent collaborative perception enhances sensing capabilities by enabling agents to share perceptual data. A key challenge lies in handling *heterogeneous* features from agents equipped with different sensing modalities or model architectures, which complicates data fusion. Existing approaches often require retraining encoders or designing interpreter modules for pairwise feature alignment, but these solutions are not scalable in practice. To address this, we propose *GT-Space*, a flexible and scalable collaborative perception framework for heterogeneous agents. GT-Space constructs a common feature space from ground-truth labels, providing a unified reference for feature alignment. With this shared space, agents only need a single adapter module to project their features, eliminating the need for pairwise interactions with other agents. Furthermore, we design a fusion network trained with contrastive losses across diverse modality combinations. Extensive experiments on simulation datasets (OPV2V and V2XSet) and a real-world dataset (RCooper) demonstrate that GT-Space consistently outperforms baselines in detection accuracy while delivering robust performance. Our code will be released at https://github.com/KingScar/GT-Space.

## 1 Introduction

Collaborative perception can enhance the sensing capabilities of interconnected vehicles through data sharing (Xu et al., 2022c; Wang et al., 2024c; Xu et al., 2022b; Wang et al., 2024b; Hao et al., 2024). The sharing is typically conducted at the feature level, where agents exchange compressed feature data rather than raw sensor data to ensure communication efficiency. The collaboration is called *homogeneous* when the shared feature data are aligned in both semantics and granularity (Yang et al., 2023a; Cui et al., 2022; Lei et al., 2022), for example through the same encoder model; otherwise, it is considered *heterogeneous*. The latter case is common in practice, as the perception units, or *agents*, often differ in sensor modality or model architecture. If not properly handled, such an issue can significantly degrade the collaboration performance of the agents.

Current solutions (Xu et al., 2022b; Xiang et al., 2023) to fuse heterogeneous feature data typically involve a feature adaptation step before performing fusion. Two main adaptation strategies are commonly used: (1) Encoder retraining (Lu et al., 2024), illustrated in Fig. 1(a). Here, the ego agent runs both a perception model and a fusion network. To align with the ego in feature space, a collaborating agent needs to retrain its encoder. In open environments, this means maintaining multiple encoders, which is highly expensive and inefficient; (2) Feature interpreter (Luo et al., 2024), shown in Fig. 1(b). In this case, the ego must be equipped with a distinct interpreter for each heterogeneous agent, which poses a scalability issue similar to the first strategy. In both cases, collaboration performance is constrained by the ego agent's model capacity. If the ego model underperforms, the features from its collaborators may yield only limited benefits.

To address these issues, we propose *GT-Space*, a flexible and efficient framework for collaborative perception. Unlike prior methods that require pairwise feature adaptation between heterogeneous

---

*Corresponding author.

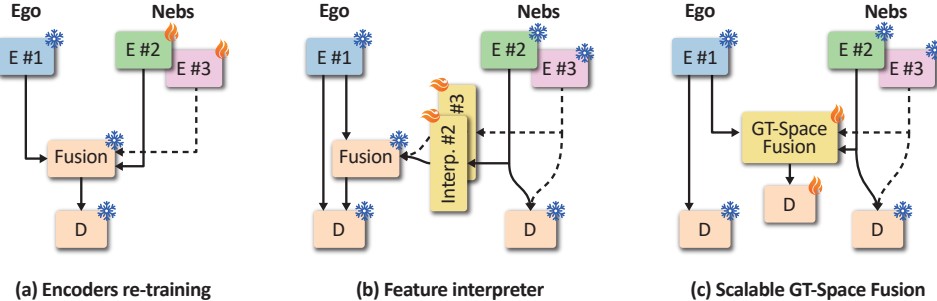

Figure 1: Comparison of heterogeneous collaborative strategies. (a) Retraining of the encoders: the fusion network is frozen while the local encoder (denoted as E) is retrained to adapt to data integration; (b) Interpreter-based alignment: local encoder and detection head (denoted as D) are frozen, the interpreter can project the received features into the ego's feature space; (c) Our method does not require any re-training of encoders or heads, and utilizes a modality-agnostic fusion module to integrate heterogeneous modalities using a ground-truth derived common feature space for alignment.

agents – forcing each agent to maintain many specialized encoders or adapters – our method needs only a single adapter model per agent. Fig. 1(c) depicts the principle of GT-Space, in which the features from individual agents are projected into a *common feature space*, and then fused for downstream object detections.

The common feature space is constructed explicitly from ground-truth object information. For each scene, the object locations, sizes, and other properties are encoded into bird's-eye view (BEV) features, which collectively form the common space. By using precise spatial and semantic information of the objects, this space provides a shared, accurate reference for aligning heterogeneous features. When a new agent joins the system, it only needs to deploy a lightweight adapter to transform its features. This design enables scalable collaboration in open environments with minimal deployment cost.

The fusion network is responsible for aggregating the adapted features from multiple agents. The ground-truth features provide strong intermediate supervision signals, guiding the fusion process more effectively than supervision from final object detection outputs. As a result, the collaboration performance is no longer limited by the weaker local model; instead, the system achieves overall improvement by exploiting complementary strengths across agents.

To ensure modality generalizability, we train the fusion network with a combinatorial loss across all modality pairs. For instance, given three models – LiDAR-based PointPillar (Lang et al., 2019), SECOND(Yan et al., 2018), and camera-based EfficientNet(Tan & Le, 2019) – the loss is computed over all three possible pairs. This strategy enables the model to fuse any combination of input modalities at inference time.

The main contributions of this paper are as follows:

- We present GT-Space, a flexible and efficient collaborative perception framework. The main novelty is a ground-truth derived common feature space for heterogeneous agents to align with. This approach greatly simplifies collaboration among agents, especially in an open environment.
- We propose to train the fusion network using combinatorial contrastive losses, so that the fusion network can take as input arbitrary combinations of modalities.
- Comprehensive experiments conducted on the simulation and real-world datasets demonstrate GT-Space's state-of-the-art performance in terms of generalization, plug-and-play functionality and robustness to under-performing agents.

## 2    RELATED WORK

**Collaborative Perception.** Collaborative perception studies how to efficiently utilize shared perceptual data to improve perception performance (Xu et al., 2022b; Huang et al., 2024; Yu et al., 2022; Shi

et al., 2022). The fusion of shared data can generally be performed in three ways: early, intermediate and late fusion (Xu et al., 2022a; Li et al., 2024; Shi et al., 2022). Intermediate fusion (Xu et al., 2022c; Fan et al., 2023) involves transmitting encoded feature data instead of the bulky raw data, and thus strikes a balance between accuracy and transmission cost. As such, it has been adopted as a major solution in existing works. V2VNet (Wang et al., 2020) uses multi-round message passing via graph neural networks to achieve better perception performance. To address communication overhead, Hu et al. (Hu et al., 2022) propose Where2comm, which enhances communication efficiency by reducing redundancy. Coopernaut (Cui et al., 2022) uses V2V sharing information to generate control policies for end-to-end autonomous driving. DiscoNet (Li et al., 2021) leverages knowledge distillation to enhance training by constraining the corresponding features to the ones from the network for early fusion. Some methods (Yu et al., 2024; Wei et al., 2024; Lei et al., 2022) consider employing multi-frame features to address data transmission interruption or latency.

**Heterogeneous Collaboration.** Heterogeneous collaboration has been a significant topic in real multi-vehicle systems. V2X-ViT (Xu et al., 2022b) utilizes a vision transformer to aggregate point cloud features from different agents, including vehicles and infrastructures. HM-ViT (Xiang et al., 2023) proposes a hetero-modal vision transformer to implement the feature alignment for heterogeneous sensor modalities including point clouds and RGB images. These end-to-end methods require training the entire model to fit particular combinations of modalities, which makes them inflexible when new modalities are introduced. HEAL (Lu et al., 2024) introduces a backward alignment training strategy, creating heterogeneous models by fixing a base pyramid fusion module and training only the encoders. However, retraining the encoders for the purpose of collaboration can potentially compromise the performance compared with the original encoders. PnPDA (Luo et al., 2024) proposes a plug-and-play domain adapter for aligning heterogeneous features without re-training encoders. However, the proposed adapter can only handle point cloud features and overlook sensor heterogeneity.

**Multi-modality.** Learning from multiple modalities has been an integral part of machine learning research (Lei et al., 2021; Xia et al., 2024; Yun et al., 2024; Arandjelovic & Zisserman, 2017). In the domain of intelligent vehicles, a typical combination of multi-modalities has been cameras and LiDARs (Zheng et al., 2024). BEVfusion (Liang et al., 2022), BEVGuide (Man et al., 2023) and RobBEV (Wang et al., 2024a) utilize Lidar-camera fusion to improve BEV perception in autonomous driving. (Li et al., 2021) proposes a method to project features of available modalities into a common space for fusion. (Zhang et al., 2023) proposes a unified learning encoder to simultaneously extract representations from multiple modalities with the same set of parameters. In this paper, we aim to develop a modality-agnostic collaborative perception method that can be easily deployed on any agent to enhance perception capabilities.

## 3 METHODOLOGY

### 3.1 BASIC PIPELINE

In the multi-agent system, each agent is equipped with its own sensors and perception models. The processing pipeline of each agent generally involves feature encoding, compression, transmission, decompression, fusion and decoding. Specifically, upon obtaining input raw sensor data $O_i$, each agent $i$ encodes the data into a BEV feature vector $F_i$. To reduce transmission time, a compressor is used to compress $F_i$ to $\hat{F}_i$ before transmitting them to the collaborative agents. As a receiver, the agent $i$ receives compressed BEV features $\hat{F}_j$ from another agent and decompresses them to $F_j$. Then the fusion network collects and aggregates all received BEV features to generate a consolidated BEV features $H_i$. Finally, the decoder processes $H_i$ to output the detections $B_i$.

### 3.2 COLLABORATION FRAMEWORK

Fig. 2 shows the pipeline of the GT-Space. For the training process, we first train the perception network of a single agent, where raw sensor inputs are encoded into BEV feature maps and processed by a detection head to produce detection results, yielding a set of trained encoders. Next, the ground truth encoder is trained to map object labels into BEV representations, which a decoder reconstructs as bounding boxes, defining the ground feature space. Finally, during the training of the fusion network, the encoder weights are kept frozen, heterogeneous features are projected into a shared space via modality-specific projectors, and the network is trained using combinations of a few agents.

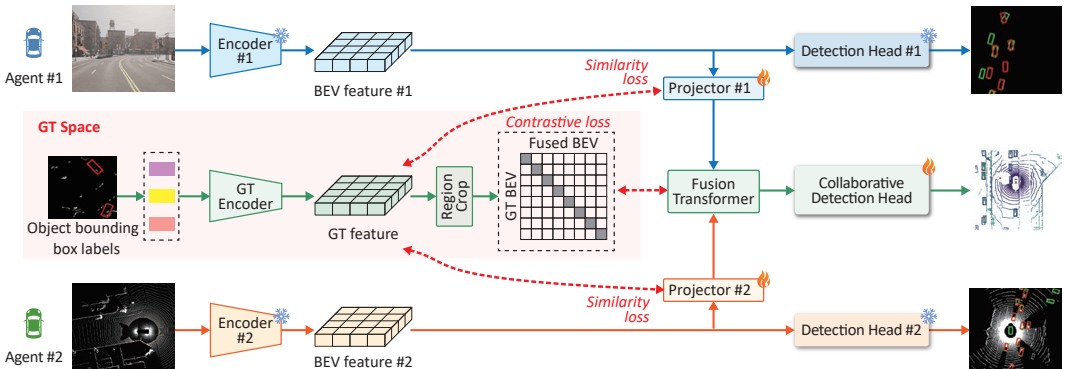

Figure 2: The framework of GT-Space. Heterogeneous features undergo domain conversion before being input into the fusion network. The ground truth object bounding box labels are leveraged to generate a common feature space. Heterogeneous features from different agents are projected into the common feature space for alignment and fusion.

During inference, each heterogeneous agent aligns its features to the ground feature space through its own projector, which are then passed through the fusion network and subsequently through the collaborative detection head to generate the final results.

When a previously unseen agent joins the collaboration, we freeze all parameters and only train the projector assigned to this agent, enabling it to adapt to the pre-trained fusion network and participate in collaborative perception.

## 3.3 COMMON FEATURE SPACE

Previous methods typically align heterogeneous features by projecting them into a fully learned latent space. The representations are learned under the supervision signals from the object detection output. In contrast, our proposed common feature space is able to introduce additional feature level supervision derived from ground truth labels. The can help with bridging domain gaps between heterogeneous agents.

As shown in Fig. 3 (a), given the ground truth bounding box annotations of the objects in the scene, we use such label information to generate BEV features. Each 3D bounding box can be formulated as a vector:

$$B_i = (x, y, z, l, w, h, r, c), \tag{1}$$

where $(x, y, z)$ is the center point, $(l, w, h)$ the length, width, and height of the 3D bounding box, respectively; $r$ the yaw angle with respect to a predefined axis, and $c$ the category of the object. Given bounding box $B_i$, we utilize two fully-connected (FC) layers with layer normalization (LN) to encode the object-relevant information,

$$\beta_i = \text{LayerNorm}(\text{FC}(B_i)), \tag{2}$$

where $\beta_i$ denotes the encoded representation.

Then we map the representations of objects onto the BEV plane to construct BEV features. A BEV feature frame is a grid map, where each grid cell of the map represents a region in the scene (Li et al., 2022; Yang et al., 2023b). A grid cell $c$ is a square area and is assigned a pair of integer coordinates $(x_c, y_c)$. Let $U_c$ be the feature of the $c$-th cell, defined as:

$$U_c = \text{MLP}(\beta_i, \text{PE}(x_c, y_c)), \tag{3}$$

where $\text{PE}(\cdot)$ denotes the Sinusoidal position embedding (Vaswani et al., 2017).

An object can cover multiple grid cells, thus the object $B_i$ comprises a set of features $S_{B_i} = \{U_1^i, ..., U_n^i\}$ where $n$ is the number of cells in $B_i$. If multiple objects cover the same grid cell $c$, we sum up $c$'s features from all the covering objects, and assign the aggregated feature to $c$. This preserves information of overlapping objects. In this way, we can obtain a BEV map $F_{GT}$ comprising all the cells' features.

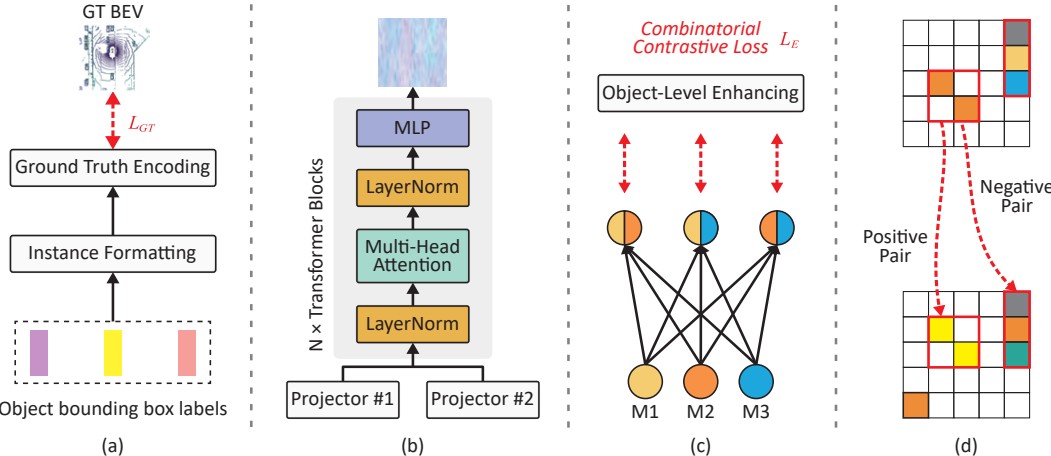

Figure 3: Feature alignment in GT-Space. (a) Ground truth features generation: The object bounding box labels are encoded and mapped onto the BEV map; (b) Multi-modal transformer: Each modality is mapped into a common feature space by a specific projector; (c) Combinatorial contrastive learning: M1, M2, and M3 denote three different modalities, and computing contrastive loss over all pairs of these modalities enhances the model's general feature extraction ability; (d) Object-level alignment: Contrastive learning pulls features of the same object closer and pushes those of different objects apart.

To ensure the effectiveness of the generated BEV feature map, following BEVFormer (Li et al., 2022), we feed the features encoded by Eq. 2 and Eq. 3 into the detection head to output detection results, and supervise the training process with the Intersection over Union (IoU) loss $L_{GT}$:

$$L_{GT} = \frac{1}{K} \sum_{k=1}^{K} \left(1 - IoU_k\right), IoU_k = |P_k \cap G_k| / |P_k \cup G_k|, \tag{4}$$

where $G_k$ denotes the ground truth bounding box, and $P_k$ represents the output of the detection head for the $k$-th object, and $K$ the total number of objects. This ensures that the encoded object-level BEV features can be decoded into bounding box outputs, thereby obtaining the ground truth BEV feature map $F_{GT}$.

### 3.4 HETEROGENEOUS FEATURE ALIGNMENT

**Heterogeneity Alignment.** For heterogeneous features, even feature elements with the same semantics may differ in dimension and parameter scales because of the domain gaps. As shown in Fig. 3 (b), we adopt a specific projector $\Phi_a$ that can project the local BEV feature into the common space as follows:

$$\Phi_a = \arg \min_{\eta} L_\eta(F_{GT}, F_a), \tag{5}$$

$$L_\eta = ||F_{GT} - \eta(F_a)||_2, \tag{6}$$

where $F_{GT}$, $F_a$ denotes the common feature map and local feature map generated by the $a$-th agent, respectively. $\eta$ denotes the feature transformation and $L_\eta$ denotes the feature similarity loss function (Wang et al., 2025).

**Relevant Feature Enhancement.** To ensure the model effectively captures critical information for detection, we expect the fusion network to concentrate on object-relevant features. We introduce a transformer model to handle arbitrary heterogeneous features. As shown in Fig. 3 (b), each block consists of a multi-head self-attention and an FC layer with two LN transforms. It should be noted that the feature enhancement is achieved through the training strategy.

We leverage contrastive learning to supervise the fusion network learning. Specifically, the goal is to align the fused features with the ground-truth features. First, we assume that modalities $m$ and $m'$ are fused, resulting in the fused feature $F_{m,m'}$. To compute the contrastive loss, the pooling operation is

used for each object to obtain the mean features to enable representation consistency. For the grid cell $c$ covered by the object bounding box $B$, we calculate the temperature-controlled cosine similarity between the local BEV feature and the ground truth feature.

$$s_{B,c,P} = \frac{(F_{m,m'}^{B,c})^\top \bar{U}^P}{\tau}, \tag{7}$$

where $F_{m,m'}^{B,c}$ denotes the feature of grid cell $c$ in fused BEV map $F_{m,m'}$ and $\bar{U}^P$ is the averaged ground feature for the object bounding box $P$, and $\tau$ is the temperature parameter. As shown in Fig. 3 (d), the cross-entropy loss is used to maximize the similarity between the fused features and ground truth features for the same object, while minimizing the similarity for different objects:

$$L_{m,m'} = -\sum_{B \in \mathcal{B}} \sum_{c \in cells(B)} \log\left(\frac{\exp(s_{B,c,B})}{\sum_{l \in \mathcal{B}} \exp(s_{B,c,l})}\right). \tag{8}$$

where $\mathcal{B}$ denotes the set of objects. Then we adopt a combinatorial contrastive to enhance the model's ability to handle different modalities. As illustrated in Fig. 3 (c), all possible modality pairs are considered as the inputs, which can be defined as:

$$L_E = \sum_{m,m'} L_{m,m'}, \tag{9}$$

where $L_{m,m'}$ denotes the contrastive loss between the fused feature of modality pair $m$ and $m'$ and the ground truth feature as in Eq. 8. This combinatorial joint optimization helps the model effectively capture the object-relevant information, which is the core of heterogeneous perception features.

## 3.5 TRAINING

During the training of our framework, the models of individual agents, including local encoders and detection heads are pre-trained and frozen during the training, thus enabling a plug-and-play functionality for easy deployment. Since our fusion objective is to achieve alignment across heterogeneous features, in order to avoid the noise caused by spatial misalignment, we train the model using the observation data from a single agent, which is spatially aligned.

We use a series of pretrained encoders to obtain heterogeneous BEV feature maps of the same scene, and use a trained ground-truth feature encoder to obtain the ground-truth BEV features for constructing the common feature space. For agent $a$, each BEV feature is fed into its corresponding projector $\Phi_a$ and transformed into the common feature space, then passed through the fusion network to produce fused features, which are finally fed into the detection head to generate detection results. Three losses are considered in the process of training: (1) feature alignment loss $L_{\Phi_a}$ (Eq. 6); (2) heterogeneous fusion loss $L_E$ to enhance the object-specific features (Eq. 9); (3) base BEV detection loss $L_B$. The overall training objective is therefore:

$$L = \sum_a L_{\Phi_a} + L_E + L_B. \tag{10}$$

The ground truth BEV features are applied exclusively during the training process via $L_\Phi$ and $L_E$ without adding additional network architectures and parameters. Thus, we can obtain a general fusion model using a few agents that can handle arbitrary modality inputs and effectively capture object-relevant features. This endows the model with scalability: when a new agent with unseen modalities joins, it only requires training its projector to adapt to the trained fusion model and enable collaboration.

## 4 EXPERIMENTS

### 4.1 DATASETS

**OPV2V** (Xu et al., 2022c). OPV2V provides LiDAR point cloud and RGB data containing multiple autonomous vehicles. Using the OpenCDA (Xu et al., 2021), a cooperative driving framework and the CARLA (Dosovitskiy et al., 2017) simulator, the dataset contains a total of 10,915 frames, which are split into train/validation/test sets with quantities of 6,765/1,980/2,170, respectively.

**V2XSet** (Xu et al., 2022b). V2XSet builds upon OPV2V by adding roadside infrastructures, thereby introducing heterogeneous V2I collaboration.

**RCooper** (Hao et al., 2024). RCooper is a large-scale roadside cooperative perception dataset with LiDAR and camera data, 3D bounding boxes, and diverse traffic scenarios, supporting multi-modal 3D detection and tracking. The results are shown in Appendix. D.

## 4.2 EXPERIMENTAL SETTINGS

**Evaluation Metrics.** We adopt 3D detection accuracy as the metric to evaluate the performance of the model, which is measured by average precision (AP) at IoU thresholds of 0.5 and 0.7. The perception area of the agents are set to $x \in [-140.8m, 140.8m]$, $y \in [-40m, 40m]$.

**Implementation details.** As shown in Table. 1, we assume four types of agents, including two LiDAR models and two camera models. Agent 2 acts as a vehicle when using the OPV2V dataset, whereas in V2XSet, it is set as an infrastructure unit. We build our model using the OpenCOOD framework (Xu et al., 2022c) and train it on all datasets using an NVIDIA A100 GPU. For the local encoders and detection heads, we follow the same hyperparameters as prior works (Luo et al., 2024; Shao et al., 2024). Local models and ground truth encoder are trained for 30 epochs using the Adam optimizer (Kingma & Ba, 2014).During the training of the fusion model, the original encoders and detection heads of all the agents are frozen, ensuring our fusion model does not affect the original perception capability of individual agents when collaborative perception is not in use.

**Compared Methods.** We compare our method with existing methods on 3D object detection. Non-collaboration is considered as our baseline method, which only use the ego's perception data. We also evaluate the Late Fusion, in which an agent transmits its detection results and the ego leverages Non-maximum suppression to generate the final predictions. For the intermediate collaborative methods, we benchmark the following approaches: End-to-end training (Xiang et al., 2023), PnPDA (Luo et al., 2024), HEAL (Lu et al., 2024), Hetercooper (Shao et al., 2024) and STAMP (Gao et al., 2025).

Table 1: Settings of heterogeneous agents in the experiments.

| Agent | Agent 1 | Agent 2 | Agent 3 | Agent 4 |
|---|---|---|---|---|
| Carrier | Vehicle | Infrastructure / Vehicle | Vehicle | Vehicle |
| Sensor | LiDAR | LiDAR | Camera | Camera |
| Model | SECOND (Yan et al., 2018) | PointPillar (Lang et al., 2019) | EfficientNet (Tan & Le, 2019) | ResNet50 (He et al., 2016) |

## 4.3 QUANTITATIVE RESULTS

**Performance of Different Modality Pairs.** We compare the detection performance of GT-Space against other collaborative methods under heterogeneous perception settings, as shown in Tab. 2. Agent 1 is fixed as the ego agent, while agents A2, A3, and A4 serve as collaborators. Across all cases, GT-Space consistently outperforms competing methods, due to the shared common feature space, which provides explicit object-level supervision for feature alignment and leads to more effective feature fusion.

As expected, LiDAR-based perception achieves higher detection accuracy than RGB-based methods, owing to the richer spatial information inherent in point clouds. Notably, our approach yields larger gains for agent pairs with higher modality heterogeneity, highlighting its ability to bridge representation gaps across different domains. Specifically, the use of contrastive learning in GT-Space enhances object-relevant features in a way that end-to-end training or feature interpreter methods cannot achieve.

**Heterogeneous Collaboration.** We further evaluate the perception performance of four heterogeneous agents, A1-A4, as listed in Tab. 3. The results on OPV2V and V2XSet are in shown in Tab. 3, The interpreter-based PnPDA and STAMP, provide substantial improvements for LiDAR agents; however, their ability to enhance camera agents remains limited. This is because spatial information in point clouds can be lost during the interpretation process and is difficult to recover with a frozen camera detection head. In contrast, GT-Space leverages the common feature space as a reliable reference for heterogeneous feature fusion, thereby achieving the best performance across all agents.

Table 2: Comparison of fusion performance for different heterogeneous modality pairs using AP@50 and AP@70 metrics. A1, A2, A3 and A4 refer to agent 1, agent 2, agent 3 and agent 4, respectively.

| Dataset | Method | $AP@50 \uparrow$ | | | $AP@70 \uparrow$ | | |
|---|---|---|---|---|---|---|---|
| | **Ego** A1, **collaborates with** | A2 | A3 | A4 | A2 | A3 | A4 |
| OPV2V | Non-collaboration | 0.738 | 0.738 | 0.738 | 0.614 | 0.614 | 0.614 |
| | Late fusion | 0.821 | 0.772 | 0.766 | 0.680 | 0.615 | 0.613 |
| | HM-ViT (Xiang et al., 2023) | 0.857 | 0.814 | 0.807 | 0.731 | 0.646 | 0.652 |
| | PnPDA (Luo et al., 2024) | 0.878 | 0.809 | 0.802 | 0.792 | 0.651 | 0.653 |
| | HEAL (Lu et al., 2024) | 0.887 | 0.827 | 0.823 | 0.801 | 0.724 | 0.728 |
| | Hetecooper (Shao et al., 2024) | 0.881 | 0.812 | 0.807 | 0.804 | 0.703 | 0.684 |
| | STAMP (Gao et al., 2025) | 0.876 | 0.833 | 0.827 | 0.806 | 0.734 | 0.738 |
| | GT-Space (ours) | **0.891** | **0.848** | **0.844** | **0.810** | **0.766** | **0.762** |
| V2XSet | Non-collaboration | 0.671 | 0.671 | 0.671 | 0.537 | 0.537 | 0.537 |
| | Late fusion | 0.719 | 0.674 | 0.667 | 0.629 | 0.541 | 0.548 |
| | HM-ViT (Xiang et al., 2023) | 0.803 | 0.741 | 0.736 | 0.710 | 0.625 | 0.631 |
| | PnPDA (Luo et al., 2024) | 0.835 | 0.749 | 0.743 | 0.763 | 0.623 | 0.634 |
| | HEAL (Lu et al., 2024) | 0.854 | 0.785 | 0.791 | 0.778 | 0.693 | 0.688 |
| | Hetecooper (Shao et al., 2024) | 0.860 | 0.759 | 0.747 | 0.782 | 0.684 | 0.680 |
| | STAMP (Gao et al., 2025) | 0.858 | 0.801 | 0.796 | 0.774 | 0.703 | 0.692 |
| | GT-Space (ours) | **0.874** | **0.826** | **0.830** | **0.802** | **0.741** | **0.738** |

Note that the improvements are more pronounced for camera agents, reflecting the strength of our approach in compensating for weaker agents.

**Robustness to Under-Performing Agents.** Agents with weaker capabilities may contribute low-quality features, which may bring down the collaboration performance. We further investigate the robustness of the system with respect to under-performing agents, as shown in Fig. 4. The weaker agents are obtained by modifying the training setups of their perception models. It can be seen that our method outperforms the baselines in collaboration, because the common feature space offers a strong reference for the agents to align with, and the centralized fusion network is sufficiently trained with full cross-modality losses. Since LiDAR agents provide more precise perception information in the collaborative system, the performance gains in heterogeneous scenarios primarily depend on LiDAR. Consequently, compared to camera agents, variations in LiDAR agents' performance lead to larger fluctuations in the overall collaborative performance.

Table 3: Perception performance of different agents in multi-agent scenarios.

| Dataset | Method | $AP@50 \uparrow$ | | | | $AP@70 \uparrow$ | | | |
|---|---|---|---|---|---|---|---|---|---|
| | | A1 | A2 | A3 | A4 | A1 | A2 | A3 | A4 |
| OPV2V | Non-collaboration | 0.738 | 0.744 | 0.402 | 0.396 | 0.614 | 0.620 | 0.354 | 0.337 |
| | Late fusion | 0.818 | 0.818 | 0.818 | 0.818 | 0.690 | 0.690 | 0.690 | 0.690 |
| | HM-ViT (Xiang et al., 2023) | 0.852 | 0.853 | 0.835 | 0.837 | 0.756 | 0.752 | 0.722 | 0.725 |
| | PnPDA (Luo et al., 2024) | 0.861 | 0.864 | 0.828 | 0.833 | 0.798 | 0.794 | 0.719 | 0.716 |
| | HEAL (Lu et al., 2024) | **0.894** | 0.889 | 0.842 | 0.843 | 0.806 | 0.801 | 0.726 | 0.733 |
| | Hetecooper (Shao et al., 2024) | 0.876 | 0.881 | 0.836 | 0.838 | 0.802 | 0.798 | 0.731 | 0.739 |
| | STAMP (Gao et al., 2025) | 0.883 | 0.886 | 0.840 | 0.831 | **0.815** | 0.801 | 0.718 | 0.716 |
| | GT-Space (ours) | 0.892 | **0.894** | **0.867** | **0.859** | 0.814 | **0.803** | **0.758** | **0.750** |
| V2XSet | Non-collaboration | 0.671 | 0.665 | 0.389 | 0.384 | 0.537 | 0.542 | 0.323 | 0.330 |
| | Late fusion | 0.798 | 0.798 | 0.798 | 0.798 | 0.628 | 0.628 | 0.628 | 0.628 |
| | HM-ViT (Xiang et al., 2023) | 0.824 | 0.818 | 0.809 | 0.811 | 0.715 | 0.713 | 0.698 | 0.692 |
| | PnPDA (Luo et al., 2024) | 0.831 | 0.822 | 0.812 | 0.806 | 0.764 | 0.752 | 0.694 | 0.697 |
| | HEAL (Lu et al., 2024) | 0.859 | 0.860 | 0.823 | 0.826 | 0.786 | 0.781 | 0.698 | 0.701 |
| | Hetecooper (Shao et al., 2024) | 0.852 | 0.849 | 0.818 | 0.816 | 0.780 | 0.781 | 0.726 | 0.724 |
| | STAMP (Gao et al., 2025) | 0.854 | 0.848 | 0.817 | 0.815 | 0.801 | 0.794 | 0.709 | 0.717 |
| | GT-Space (ours) | **0.873** | **0.866** | **0.845** | **0.848** | **0.806** | **0.804** | **0.742** | **0.733** |

**Robustness to Imperfect Localization.** Existing experiments typically assume that each agent has access to an accurate pose. However, in real-world scenarios, localization noise is inevitable, leading to imperfect spatial alignment across agents. To evaluate robustness under such conditions,

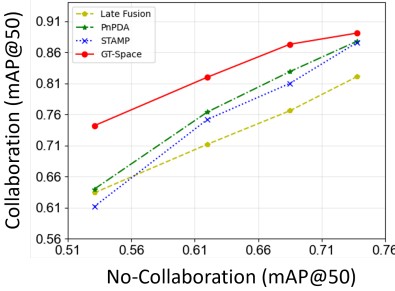 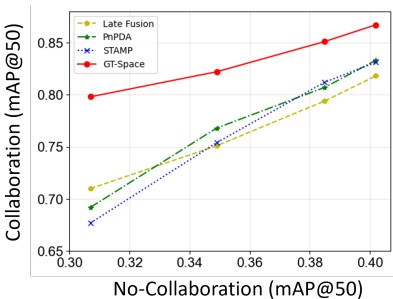

Figure 4: Performance of under-performing agents on OPV2V. The left column represents Agent 1, and the right column Agent 3. The horizontal axis represents performance without collaboration, while the vertical axis represents collaboration.

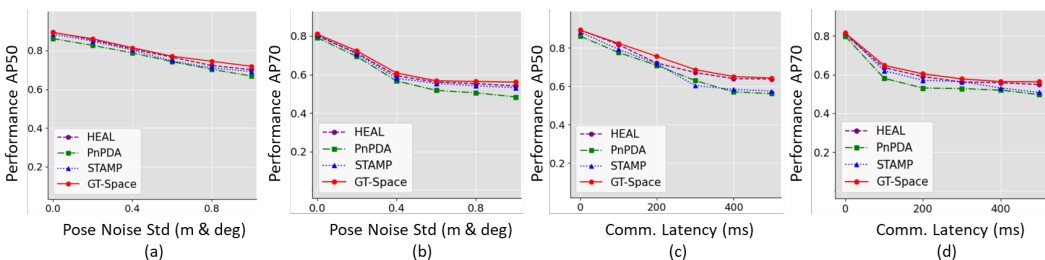

Figure 5: Robust Experiment to pose error and communication latency.

we introduce a pose-error experiment by adding Gaussian noise into the ground-truth poses, where the pose noise is set to $N(0, \sigma_p^2)$ on x,y location and $N(0, \sigma_r^2)$ on yaw angle (Lu et al., 2023). A1 is set as the ego agent and the performance is shown in Fig. 5. The results demonstrate that our method consistently maintains state-of-the-art performance across a wide range of pose error conditions.

**Robustness to Communication Latency.** In real-world scenarios, communication latency is inevitable. We further evaluate the robustness of our method under different latency conditions. Since the sensor data in the OPV2V dataset is recorded at 10 Hz, we simulate a communication latency of $k \times 100$ ms by replacing the first $k$ transmitted feature frames with the current frame for each collaborative sample, thereby emulating the latency effect. Following (Yu et al., 2024), we set the range of communication latency to 100–500 milliseconds. Results in Fig. 5 demonstrate that even with a 500 ms latency, our method still outperforms the baseline methods.

## 4.4 ABLATION STUDIES

To investigate the effectiveness of each component in our system, we introduce four variants of our GT-Space: (1) w/o-GT features, which replaces the ground-truth feature space with a unified feature space generated by PointPillar; (2) w/o-projector, which excludes the feature projector, i.e., feed the heterogeneous features directly into the transformer for fusion; (3) w/o-combinatorial contrastive loss, which trains the fusion network solely with the basic detection loss, without the proposed combinatorial contrastive losses. Tab. 4 reports the performance of Agent 1 under these design variants. The results show that removing the feature projector causes the largest performance degradation. This is because heterogeneous features have distinct semantic representations, making it difficult for the fusion network to effectively distinguish and aggregate them without proper alignment. For Agent 1, a LiDAR agent, replacing the GT feature space with the point cloud feature space still yields reasonably good performance, as the point

Table 4: Effect of each design component in the model.

| Configurations | OPV2V | | V2XSet | |
|---|---|---|---|---|
| | mAP@50 | mAP@70 | mAP@50 | mAP@70 |
| w/o-GT feature | 0.868 | 0.795 | 0.830 | 0.782 |
| w/o-projector | 0.791 | 0.683 | 0.716 | 0.604 |
| w/o-contrastive loss | 0.845 | 0.721 | 0.823 | 0.709 |
| Full version | **0.892** | **0.814** | **0.873** | **0.806** |

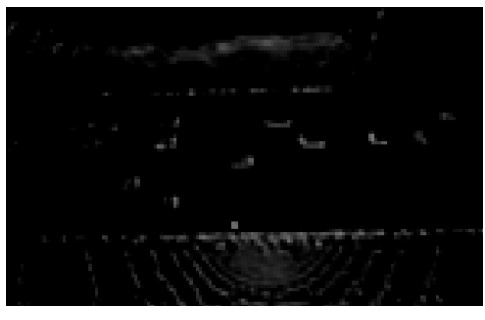

(a) Feature map before fusion             (b) Feature map after fusion

Figure 6: Visualization of BEV features before and after fusion. (a) Features before fusion of A2 (PointPillar); (b) Enhanced features after fusion.

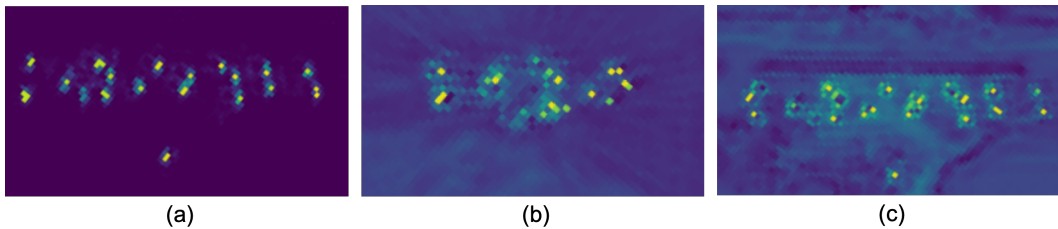

(a)             (b)             (c)

Figure 7: Visualization of (a) Ground truth feature, (b) projected camera feature $\Phi_{cam}(F_{cam})$ and (c) projected LiDAR feature $\Phi_{lidar}(F_{lidar})$.

cloud representation inherently preserves rich geometric cues. Finally, removing the combinatorial contrastive loss results in a noticeable decline in detection accuracy, since the contrastive objective not only enforces cross-modality alignment but also strengthens feature representations.

### 4.5 VISUALIZATION

Fig. 6 presents the intermediate feature maps of A2 before and after data fusion. A clear contrast can be observed: prior to fusion, the object-relevant features exhibit weak activations and are heavily contaminated by noise. After fusion, however, the feature map highlights significantly more object information, confirming the effectiveness of collaborative perception, while the noise is notably suppressed, reflecting the role of ground-truth features as strong supervision signals.

We further present the visualization of ground truth feature, projected camera feature $\Phi_{Cam}(F_{Cam})$ and projected LiDAR feature $\Phi_{LiDAR}(F_{LiDAR})$. As shown in Fig. 7, we can observe that the ground truth feature map contains only object-related features. Since point cloud features include richer spatial information, the mapped $F_{\text{LiDAR}}$ still retains some road information. As for the camera features $F_{\text{Cam}}$, note that the projector does not have the ability to enhance features; therefore, the object-related features in the feature map are not as abundant as in $F_{\text{LiDAR}}$.

## 5 CONCLUSION

In this paper, we have presented GT-Space, a scalable and high-accuracy collaborative perception method aimed at fusing shared data among agents that are heterogeneous in sensor modality or data encoders. Ground truth object labels are used to construct a common feature space for feature alignment and fusion. To enable the fusion network to handle arbitrary input modalities and enhance relevant features, we employ a combinatorial contrastive loss for training. Experiments on the OPV2V, V2XSet and RCooper datasets validate the effectiveness of GT-Space. GT-Space depends on ground-truth annotations and ideal communication/pose conditions. Future work will focus on weakly-supervised learning to enhance its real-world applicability.

## 6 ACKNOWLEDGMENTS

This work was supported by Shenzhen Natural Science Foundation under grant 202412023000612.

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

## A  COLLABORATIVE PERCEPTION

A collaborative perception system consists of multiple agents, each equipped with its own sensors and perception models. The collaboration model is generally composed of an encoder, a compressor, a decompressor, a feature fusion module and a decoder, where $i$ denotes the agent.

The process of the $i$-th agent can be formally represented as follows:

Feature Encoding: $$F_i = E_i(O_i), \tag{1a}$$

Compression: $$\hat{F}_i = \phi_i(F_i), \tag{1b}$$

Transmission: $$\hat{F}_{j \to i} = \iota_{j \to i}(\hat{F}_j), j \in \mathcal{S}, \tag{1c}$$

Decompression: $$F_j = \phi_i^{-1}(\{\hat{F}_{j \to i}\}_{j \in \mathcal{S}}), \tag{1d}$$

Fusion: $$H_i = \psi_i(\{F_j\}_{j \in \mathcal{S}}), \tag{1e}$$

Decoding: $$B_i = D_i(H_i), \tag{1f}$$

where $E_i$ is the encoder, $\phi_i$ is compressor, $\iota$ is the communication operator, $\psi$ is the fusion module, and $H_i$ is the aggreagated feature and $B_i$ is the final output obtained by the detection decoder $D_i$.

## B  SUMMARY OF SYMBOLS

We summarize all the relevant symbols in Tab. 5.

Table 5: Summary of Symbols.

| Symbol | Meaning |
|---|---|
| $F_a$ | Feature map from $a$-th agent. |
| $F_{GT}$ | Ground truth feature map. |
| $B_i$ | $i$-th bounding box in a frame. |
| $\beta_i$ | Encoded representation of $B_i$ |
| $U_c$ | The feature of the $c$-th cell |
| $U_n^i$ | The feature of the $n$-th cell of box $B_i$ |
| $S_{B_i}$ | The set of features of box $B_i$ |
| $L_{GT}$ | Intersection over Union (IoU) loss |
| $L_\eta$ | Feature similarity loss function. |
| $\Phi_a$ | Feature projector for the $a$-th agent. |
| $F_{m,m'}$ | The fused feature of modality $m$ and $m'$. |
| $F_{m,m'}^{B,c}$ | The feature of grid cell $c$ in fused BEV map $F_{m,m'}$. |
| $\tau$ | Temperature parameter. |
| $\bar{U}^P$ | The averaged ground feature for the object bounding box $P$. |

## C  IMPLEMENTATION DETAILS

**Compared Methods.** GT-Space is compared with seven baselines:

- Non-collaboration Method;
- Late fusion which transmits detected results;
- HM-ViT (Xiang et al., 2023) based on different encoders using end-to-end training;
- PnPDA (Luo et al., 2024). PnPDA adopts modality interpreter to transform heterogeneous features to a semantic space for fusion.
- HEAL (Lu et al., 2024). HEAL firstly trains a base collaboration fusion network with frozen parameters and re-trains the encoder for generalized fusion.
- Hetecooper (Shao et al., 2024). Hetecooper constructs a graph transformer to achieve heterogeneous collaboration.

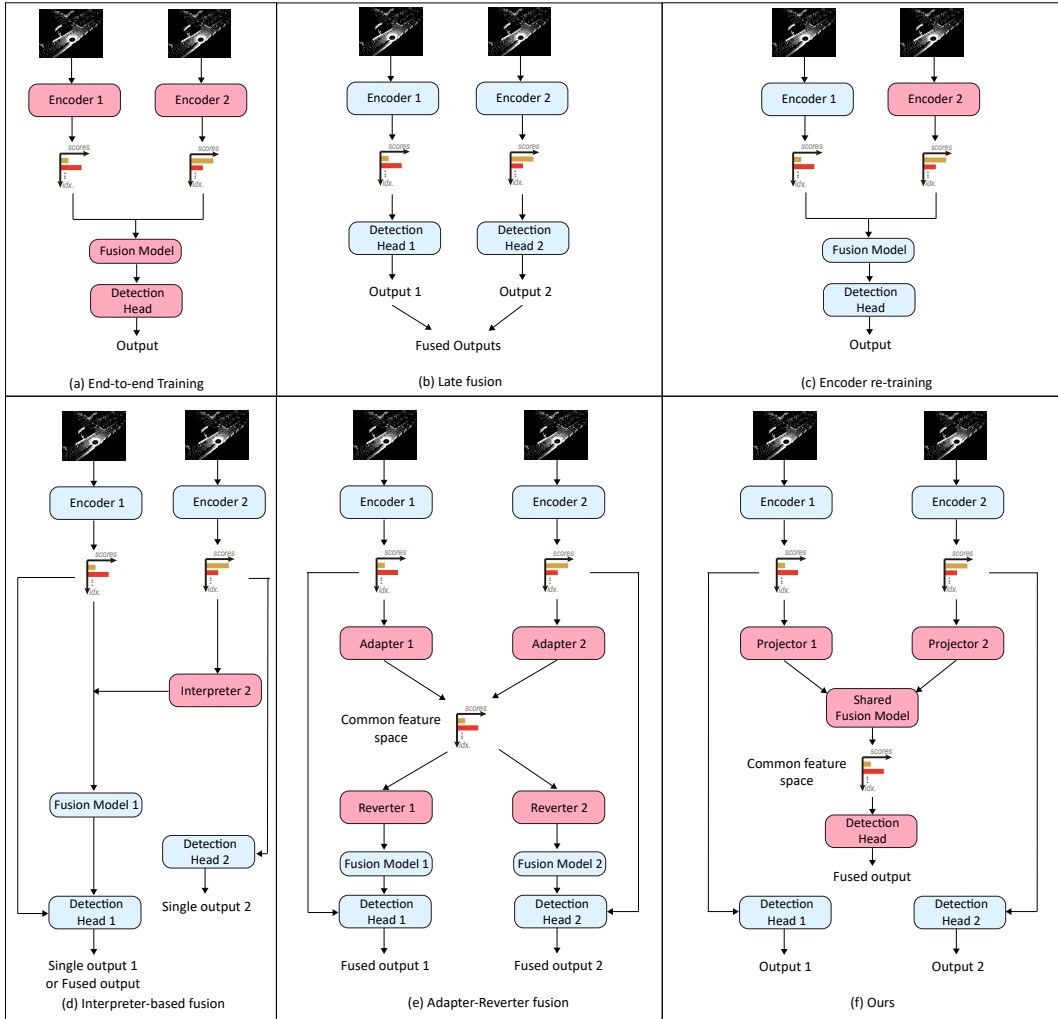

Figure 8: (a) End-to-end training; (b) Late fusion; (c) Encoder re-training; (d) Interpreter-based method; (e)Adapter-Reverter method; (f) GT-Space(Ours). Blue indicates frozen modules, while red represents modules that are updated during training.

- STAMP (Gao et al., 2025). STAMP configures an adapter and a reverter for each modality, enabling flexible transformation from local modality features to protocol features.

**Architectures of Different Methods.** Fig. 8 shows various heterogeneous collaborative frameworks. Late fusion simply transmits and combines agent detected results after local processing. HM-ViT (Xiang et al., 2023) adopts end-to-end training, which is effective but interferes with individual perception. HEAL (Lu et al., 2024) employs the fixed detection decoder and fusion model and re-trains only the local encoders, but still faces the issue of the coexistence of individual perception and collaborative perception. Interpreter-based methods adopt modality-specific interpreters to transform heterogeneous features into a common semantic feature space for alignment. The Adapter-Reverter method is a variant of interpreter-based approaches. It first uses an adapter to project local features into a common feature space, and then employs a reverter to map the common feature space back to the local features.

**Projector Architecture.** We use the same architectures that is an MLP for projectors across all heterogeneous agents. The dimension of the broadcasting feature is set as $(128, 128, 64)$. The input dimension of projectors varies according to the feature dimension of different local models. We utilize a NVIDIA A100 GPU for training. All local models are trained on the OPV2V and V2XSet for 30 epochs, with a learning rate of 0.001 and weight decay of 0.01.

# D    ADDITIONAL RESULTS

**System Delays.** To demonstrate the deployability of our proposed collaborative system, we test the system delays of different components of the model.

Tab. 6 presents the latency breakdown of different components in the system, revealing that the majority of the computational cost is attributed to the encoding of raw perception data. In contrast, both the fusion network and the detection head consume relatively little computational resources. It is also worth noting that our method achieves better performance without significantly increasing the computational cost. This improvement stems from our enhanced training strategy, rather than the use of more complex model architectures, which would have resulted in increased parameters.

Table 6: System delays

| Method | Encoding (ms) | Fusion (ms) | Detection (ms) |
|---|---|---|---|
| HM-ViT (Xiang et al., 2023) | 7.08 | 1.38 | 0.95 |
| PnPDA (Luo et al., 2024) | 7.08 | 1.41 | 0.98 |
| Hetecooper (Shao et al., 2024) | 7.08 | 1.47 | 0.97 |
| HEAL (Lu et al., 2024) | 7.08 | 1.54 | 0.95 |
| STAMP (Gao et al., 2025) | 7.08 | 1.52 | 0.94 |
| GT-Space | 7.08 | 1.48 | 0.97 |

**Experiments on Real-world Dataset.** To further evaluate the performance of our proposed GT-Space, we conduct experiments on RCooper (Hao et al., 2024), a real-world dataset. Tab. 7 presents the performance of our approach on RCooper (AP@50), where the baselines are the same as those in Tab. 1. The scenario is an intersection and all 4 agents are roadside infrastructures. It can be seen that on the real-world dataset, our method still achieves the state-of-art performance, especially showing better results for camera agents with weaker perception capabilities.

Table 7: Performance on RCooper

| Method | A1 | A2 | A3 | A4 |
|---|---|---|---|---|
| No fusion | 0.441 | 0.438 | 0.231 | 0.228 |
| HM-ViT (Xiang et al., 2023) | 0.458 | 0.467 | 0.294 | 0.287 |
| PnPDA (Luo et al., 2024) | 0.463 | 0.468 | 0.321 | 0.330 |
| Hetecooper (Shao et al., 2024) | 0.471 | 0.465 | 0.319 | 0.324 |
| HEAL (Lu et al., 2024) | **0.478** | 0.473 | 0.343 | 0.346 |
| STAMP (Gao et al., 2025) | 0.473 | 0.469 | 0.338 | 0.342 |
| GT-Space | 0.477 | **0.475** | **0.349** | **0.351** |

**Experiments on Tracking Task.** We further evaluat the performance of our framework on the tracking task. Specifically, we conduct experiments on the RCooper (Hao et al., 2024) dataset, which adopts AB3Dmot tracker (Weng et al., 2020b) to perform object tracking based on the detection results. We adopt two evaluation metrics in (Weng et al., 2020a), 1) average multi-object tracking accuracy (AMOTA) and 2) average multi-object tracking precision (AMOTP). As shown in Tab. 8, we can see that our method outperforms all baseline models, which is consistent with the 3D detection results.

**Newly Added Agent.** Our proposed framework can adapt to newly added, previously unseen agents, but this requires a specific training strategy. Specifically, we still adopt the pipeline shown in Fig. 2. For the fusion network and the collaborative detection head, we freeze their parameters. At the same time, we also keep the encoder parameters of the newly added agent frozen, and train only its specific projector. The loss function can be expressed as:

$$L = L_{\Phi_i} + L_B,$$

where $\Phi_i$ is the projector for the newly agent $i$, which is used to project its local feature to the common feature space and $L_{\Phi_i}$ is the same as Eqa.6. $L_B$ is the base detection loss. Tab. 9 shows the performance of A1 when the collaborative system adds new agents A3 and A4.

Table 8: Performance of tracking task on RCooper

| Method | AMOTA | AMOTP |
|---|---|---|
| No fusion | 0.083 | 0.227 |
| HM-ViT (Xiang et al., 2023) | 0.224 | 0.354 |
| PnPDA (Luo et al., 2024) | 0.228 | 0.357 |
| Hetecooper (Shao et al., 2024) | 0.227 | 0.361 |
| HEAL (Lu et al., 2024) | 0.232 | 0.364 |
| STAMP (Gao et al., 2025) | 0.231 | 0.359 |
| GT-Space | 0.236 | 0.367 |

We can see that retraining-based HEAL clearly outperforms interpreter-based PnPDA and Adapter-Reverter-based STAMP, but at the cost of impairing the newly added agent's local perception. Although the projector we use also serves a modality conversion function, it does not convert inputs between agent pairs. Instead, it maps them into the shared common space. Meanwhile, the fusion network effectively enhances object-relevant representations, thereby preventing the loss of critical information.

Table 9: Newly adding agents.

| Method (Based on A1, A2, Add) | Base | + A3 | + A4 |
|---|---|---|---|
| PnPDA (Luo et al., 2024) | 0.878 | 0.857 | 0.864 |
| HEAL (Lu et al., 2024) | 0.887 | 0.891 | 0.895 |
| STAMP (Gao et al., 2025) | 0.876 | 0.863 | 0.862 |
| GT-Space | **0.891** | **0.892** | **0.897** |

**Efficiency Comparison.** To evaluate the training efficiency for the newly added agents, we additionally include four agents: A5, A6, A7, and A8. Tab. 10 reports the sensors, encoders, and encoder parameters for agents A1–A8. PointPillar (large) and SECOND (large) are obtained by increasing the number of hidden-layer parameters of the original models.

We then conduct an efficiency comparison between our method and baselines. Fig. 9 shows the number of training parameters and estimated training GPU hours. It can be seen that the retraining-based HEAL incurs high training costs. The interpreter-based PnPDA and the Adapter–Reverter–based STAMP significantly reduce the cost, though with some performance degradation. Together with Tab. 9, retraining-based HEAL achieve good performance but at high cost, while the interpreter-based method have lower cost but yield weaker results. In contrast, our proposed framework achieves strong performance with low cost.

Table 10:  Settings of heterogeneous agents in the experiments.

| Agent | Agent 1 | Agent 2 | Agent 3 | Agent 4 |
|---|---|---|---|---|
| Carrier | Vehicle | Infrastructure / Vehicle | Vehicle | Vehicle |
| Sensor | LiDAR | LiDAR | Camera | Camera |
| Model | SECOND (Yan et al., 2018) | PointPillar (Lang et al., 2019) | EfficientNet (Tan & Le, 2019) | ResNet50 (He et al., 2016) |
| Encoder Param. (M) | 3.79 | 0.87 | 56.85 | 6.88 |

| Agent | Agent 5 | Agent 6 | Agent 7 | Agent 8 |
|---|---|---|---|---|
| Carrier | Vehicle | Vehicle | Vehicle | Vehicle |
| Sensor | LiDAR | LiDAR | LiDAR | Camera |
| Model | VoxelNet (Zhou & Tuzel, 2018) | PointPillar (Large) (Lang et al., 2019) | SECOND (large) (Yan et al., 2018) | ResNet34 (He et al., 2016) |
| Encoder Param. (M) | 2.13 | 1.91 | 4.82 | 6.51 |

**Robustness to Data Distribution Shifts.** To validate the robustness to data distribution shifts, we further evaluated the fusion model trained using OPV2V on the V2X-Set dataset. As shown in Tab. 11, our method consistently outperforms the baseline models even under distribution shifts, demonstrating the robustness of the ground-truth features.

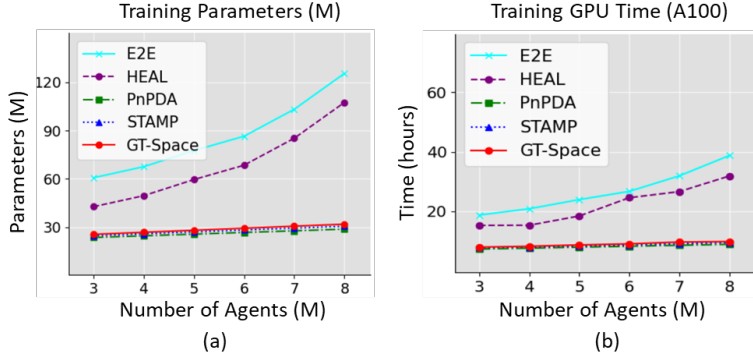

Figure 9: Training efficiency comparison for the different number of newly adding agents.

Table 11:  Performance (AP50) of agent A1-A4 in V2XSet using the fusion model trained on OPV2V.

| Dataset | Method | A1 | A2 | A3 | A4 |
|---|---|---|---|---|---|
| V2XSet | Late fusion | 0.798 | 0.798 | 0.798 | 0.798 |
| | HM-ViT (Xiang et al., 2023) | 0.809 | 0.793 | 0.792 | 0.788 |
| | PnPDA (Luo et al., 2024) | 0.819 | 0.804 | 0.801 | 0.790 |
| | HEAL (Lu et al., 2024) | 0.844 | 0.842 | 0.818 | 0.822 |
| | Hetecooper (Shao et al., 2024) | 0.835 | 0.821 | 0.810 | 0.803 |
| | STAMP (Gao et al., 2025) | 0.842 | 0.829 | 0.801 | 0.803 |
| | GT-Space (ours) | **0.858** | **0.846** | **0.828** | **0.833** |

Table 12:  Performance (AP50) of models trained with different temperature parameters

| Dataset | Temperature | A1 | A2 | A3 | A4 |
|---|---|---|---|---|---|
| OPV2V | 0.05 | 0.845 | 0.835 | 0.830 | 0.826 |
| | 0.07 | 0.850 | 0.842 | 0.824 | 0.830 |
| | 0.1 | 0.858 | 0.846 | 0.828 | 0.833 |

**Sensitivity Analysis for Temperature Parameter.** In contrastive learning, the temperature is used to adjust the "sharpness" of the similarity distribution, thereby controlling how strongly the model distinguishes between positive and negative samples. We evaluate the model's performance under different temperature settings. Based on empirical practice in multimodal tasks, the temperature parameter is typically set within the range of 0.05–0.1 (Shvetsova et al., 2022). Therefore, we evaluate three settings: 0.05, 0.07, and 0.1. The results in Tab. 12 show that a temperature of 0.1 yields the best performance. This is because smaller temperatures make the distribution sharper and strengthen the separation between positive and negative samples, but lead to training instability. Hence, 0.1 serves as a more optimal choice.

**Scale Balancing for obejcts.** Considering that large objects occupy greater weight in the loss function, we perform scale balancing accordingly. Specifically, we take the average based on the number of objects, and then multiply it by a weighting factor, which is set to the average number of objects per frame across all training frames. The modified formula of Eq. 8 is as follows:

$$L_{m,m'} = -\frac{\mu}{|\mathcal{B}|} \sum_{B \in \mathcal{B}} \sum_{c \in cells(B)} \log\left(\frac{\exp(s_{B,c,B})}{\sum_{l \in B} \exp(s_{B,c,l})}\right). \tag{12}$$

where $|\mathcal{B}|$ denotes the number of objects in a feature frame and $\mu$ is the average number of objects per frame across all training frames. We conduct experiments on the RCooper dataset, and the results are shown in the Tab. 13. It can be seen that the performance improves.

Table 13: Results of scale balancing

| Method | A1 | A2 | A3 | A4 |
|---|---|---|---|---|
| GT-Space | 0.477 | 0.475 | 0.349 | 0.351 |
| GT-Space (average) | 0.481 | 0.482 | 0.350 | 0.354 |

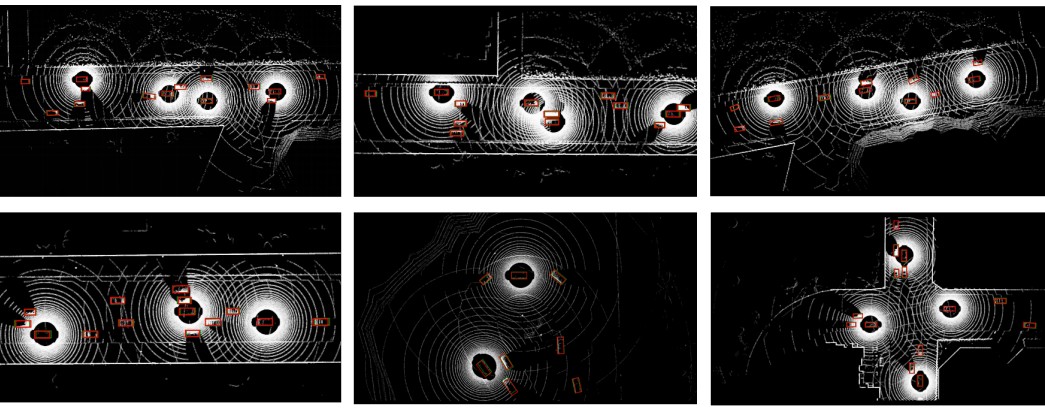

Figure 10: Visualization of collaborative perception results. We color the predicted and GT boxes.

**Additional Visualization.** Fig. 10 shows the visualization of collaborative perception results on OPV2V.

