# OpenReview forum: "GT-Space: Enhancing Heterogeneous Collaborative Perception with Ground Truth Feature Space"
_ICLR.cc/2026/Conference — ICLR 2026 Poster_

### Official Review · Reviewer_w8N3 · 2025-10-20

**Soundness:** 3
**Presentation:** 2
**Contribution:** 2
**Rating:** 4
**Confidence:** 3

**Summary:**

The paper proposes a heterogeneous collaborative perception framework named GT-Space. Instead of retraining or creating an "interpreter" for each heterogeneous agent, this framework projects the BEV (Bird's-Eye View) features from each vehicle's agent into a common feature space constructed from the ground truth, followed by fusion and detection.

**Strengths:**

- The paper has a clear motivation, aiming to solve the problem of needing to retrain for each new agent, thereby increasing the system's scalability.
- The proposed method is designed to be plug-and-play, is orthogonal to other existing methods, and can be used in conjunction with them.
- The experimental results demonstrate that the framework can effectively improve perception performance in a heterogeneous environment.

**Weaknesses:**

- The framework heavily relies on the "ground-truth feature space". The construction of GT-Space depends on ground truth. During inference, since ground truth is unavailable, the model can only "imagine" a space that aligns with it. This requires the training and deployment distributions to be sufficiently close; otherwise, the alignment will drift. The paper does not quantify the robustness under such distribution shifts.
- The authors freeze the local encoders and detection heads, promoting a "plug-and-play" capability. However, this also limits the potential performance ceiling that could be achieved through end-to-end joint fine-tuning.
- The authors claim that when adding a new, unseen agent, only the projector for that agent needs to be trained, thus enabling scalable collaboration with minimal deployment cost by deploying only a lightweight adapter. However, the paper does not explicitly report the comparison of training computation/time for "adding a new agent"; it only reports inference latency.
- The implementation process is not clearly described, containing some errors or confusing points. The lack of open-sourced code makes the paper difficult to reproduce.

**Questions:**

- The object-level contrastive learning "aggregates grids within a bounding box and then computes similarity/cross-entropy." Larger objects, which cover more grids, might have a greater weight in the loss function. Is scale balancing or re-weighting necessary?
- In Figure 1, the caption explicitly states, "Interpreter-based alignment: local encoder and detection head (denoted as D) are frozen." However, diagram (b) shows that the local encoder is trainable. Furthermore, why do agents E#2 and E#3 point to the same detection head D?
- In Figure 2, why is the similarity loss calculated between the GT feature and the BEV feature? The BEV feature is generated by a frozen encoder, so is calculating a loss on it meaningful? Shouldn't the loss be calculated on the feature after it has passed through the projector?
- Equations (5) and (6) feel somewhat vague and lack rigor. I can understand the author's intention, but the formulation seems to have a circular definition. $\Phi_{a}$ is presented as the function to be solved in the first equation, yet it is used as a known function within the loss in the second equation.
- In Equation (8), is there a missing negative sign? In $L_{m,m'} = \sum_{B} \sum_{c} \sum_{B'} \log\left( \frac{\exp(s_{B,c,B'})}{\sum_{l} \exp(s_{B,c,B'})} \right)$, is the summation in the denominator performed over all $l$? However, the term inside the exponential function being summed is $\exp(s_{B,c,B'})$, which uses the index $B'$.  Perhaps the correct formula is something like $L_{m,m'} = -\sum_{B} \sum_{c \in \text{cells}(B)} \log \frac{\exp\left(s_{B,c,B}\right)}{\sum_{l \in \mathcal{B}} \exp\left(s_{B,c,l}\right)}$.

**Details Of Ethics Concerns:**

- The citation format in the paper is clearly inconsistent. For example, in Table 2, it is evident that some papers have year information while others do not. It is recommended that the authors re-check and unify the citation format.
- It is recommended that the authors provide a detailed overview of the GT-Space framework in Section 3.2. This should include a comprehensive explanation of the workflows for pre-training, training, inference, and the process of integrating a new agent. The input, specific operations, output, and the meaning of each stage should be clearly articulated.
- It is recommended that the authors provide more detailed explanations and descriptions for the formulas and symbols used in the paper, especially when new symbols are introduced. If the code is not open-sourced, the formulas in the paper must be sufficiently comprehensive to ensure that readers can understand the method.
- It is suggested to add a comparison of system complexity in the "Newly Added Agent" section.
- It is suggested to include a sensitivity analysis for parameters such as the temperature $\tau$.

---

> ### Author Response · Authors · 2025-11-27
> **Response to Reviewer w8N3 [1/3]**
>
> We gratefully appreciate your detailed suggestions for our work. We provide the response to your concerns point by point as follows:
>
> **W1: The framework heavily relies on the "ground-truth feature space". The construction of GT-Space depends on ground truth. During inference, since ground truth is unavailable, the model can only "imagine" a space that aligns with it. This requires the training and deployment distributions to be sufficiently close; otherwise, the alignment will drift. The paper does not quantify the robustness under such distribution shifts.**
>
>
> We understand your concerns. Ground-truth information has a significant impact on performance, and overfitting to a specific distribution may result in performance degradation.
> However, we respectfully argue that this issue is effectively mitigated in our framework.
> Since the pretrained encoder is reliable, has been empirically validated, and keeps its weights frozen during the fusion network training, the projector and fusion network only need to extract the object-relevant features from the BEV map to ensure effective alignment.
>
> In addition, the GT encoder is also pretrained before the fusion network stage and does not update its weights jointly with it, which further helps prevent overfitting.
>
> We further evaluate the performance of the fusion model trained on OPV2V on the V2XSet dataset, and the results are presented below.
>
> | **Dataset** | **Method** | **A1** | **A2** | **A3** | **A4** |
> |-------------|------------|--------|--------|--------|--------|
> | **V2XSet** | Late fusion | 0.798 | 0.798 | 0.798 | 0.798 |
> |  | HM-ViT | 0.809 | 0.793 | 0.792 | 0.788 |
> |  | PnPDA | 0.819 | 0.804 | 0.801 | 0.790 |
> |  | HEAL | 0.844 | 0.842 | 0.818 | 0.822 |
> |  | Hetecooper | 0.835 | 0.821 | 0.810 | 0.803 |
> |  | STAMP | 0.842 | 0.829 | 0.801 | 0.803 |
> |  | **GT-Space (ours)** | **0.858** | **0.846** | **0.828** | **0.833** |
>
> These findings corroborate our assertion that the model does not suffer from a substantial performance degradation despite the distribution shifts.
>
> **W2: The authors freeze the local encoders and detection heads, promoting a "plug-and-play" capability. However, this also limits the potential performance ceiling that could be achieved through end-to-end joint fine-tuning.**
>
> In theory, end-to-end joint fine-tuning yields the best performance under identical conditions. However, as discussed in the paper, this approach inevitably affects the vehicle’s local perception. In scenarios without collaborating vehicles, where the system must rely solely on its own sensing capabilities, parameter modifications to the encoder can lead to degraded performance. Moreover, the training cost of such an end-to-end scheme would be substantially higher.
> Therefore, our design essentially makes a trade-off between collaborative perception and standalone perception, thereby enabling a “plug-and-play” capability.
>
> **W3: The authors claim that when adding a new, unseen agent, only the projector for that agent needs to be trained, thus enabling scalable collaboration with minimal deployment cost by deploying only a lightweight adapter. However, the paper does not explicitly report the comparison of training computation/time for "adding a new agent"; it only reports inference latency.**
>
> Thanks for your useful suggestion. We further illustrate the training computation for the scenario of adding new agents.
> The results are shown in Fig. 9 in the Appendix of the updated version.
> It can be seen that the retraining-based HEAL incurs high training costs. The interpreter-based PnPDA and the Adapter–Reverter–based STAMP significantly reduce the cost, though with some performance degradation. Together with Tab.9, retraining-based HEAL achieves good performance but at a high cost, while the interpreter-based method has a lower cost but yields weaker results. In contrast, our proposed framework achieves strong performance with low cost.
>
> **W4: The implementation process is not clearly described, containing some errors or confusing points. The lack of open-sourced code makes the paper difficult to reproduce.**
>
> Thanks for your suggestion, we have corrected the pointed errors and updated the version of the paper.  Due to the extensive use of notation, the resulting symbolic system has become rather complex. Accordingly, all symbols have been consolidated in Tab. 5 of the appendix. Then we have added further implementation details of the proposed method, and we commit to open-sourcing all code, including that for training, evaluation, and data processing.

---

> ### Author Response · Authors · 2025-11-27
> **Response to Reviewer w8N3 [2/3]**
>
> **Q1: The object-level contrastive learning "aggregates grids within a bounding box and then computes similarity/cross-entropy." Larger objects, which cover more grids, might have a greater weight in the loss function. Is scale balancing or re-weighting necessary?**
>
>
> Thank you for your insightful comment. Our design does indeed lead to such weight imbalance. Fortunately, we notice that in the OPV2V and V2XSet datasets, there are very few extremely large objects, so this imbalance does not have a significant impact. However, in the RCooper dataset, there are many objects with large size differences, such as trucks and small cars.
> Therefore, we take the average based on the number of objects, and then multiply it by a weighting factor, which is set to the average number of objects per frame across all training frames. The modified formula is as follows:
>
> \begin{equation}
>     L_{m,m'} = - \frac{\mu}{|\mathcal{B}|} \textstyle \sum_{B \in \mathcal{B}} \sum_{c \in cells(B)} \mbox{log}\Big(\frac{\mbox{exp}(s_{B,c,B})}{\sum_{l \in \mathcal{B} } \mbox{exp} (s_{B,c,l})}\Big).
> \end{equation}
>
> where $|\mathcal{B}|$ denotes the number of objects in a feature frame and $\mu$ is the average number of objects per frame across all training frames.
>
> We conduct experiments on the RCooper dataset, and the results are shown in the table below. It can be seen that the performance improves, which verifies this issue. Thank you for your suggestion, which helps improve the paper.
>
> | Method       | A1    | A2    | A3    | A4    |
> |-------------|-------|-------|-------|-------|
> | GT-Space    | 0.477 | 0.475 | 0.349 | 0.351 |
> | GT-Space (average) | 0.481 | 0.482 | 0.350 | 0.354 |
>
> **Q2: In Figure 1, the caption explicitly states, "Interpreter-based alignment: local encoder and detection head (denoted as D) are frozen." However, diagram (b) shows that the local encoder is trainable. Furthermore, why do agents E#2 and E#3 point to the same detection head D?**
>
> E2 and E3 actually use their respective detection heads. We had omitted the detection head of E3, which caused this confusion. To address this, we have corrected the figure in the updated version.
>
> **Q3: In Figure 2, why is the similarity loss calculated between the GT feature and the BEV feature? The BEV feature is generated by a frozen encoder, so is calculating a loss on it meaningful? Shouldn't the loss be calculated on the feature after it has passed through the projector?**
>
> Your understanding is correct. We indeed intend to express that the similarity loss is computed between the mapped BEV and the ground truth space. The loss is calculated on the projector.
> Therefore, we have adjusted the position of the “similarity loss” in the figure to avoid ambiguity.
>
> **Q4: Equations (5) and (6) feel somewhat vague and lack rigor. I can understand the author's intention, but the formulation seems to have a circular definition. $\Phi_{a}$ is presented as the function to be solved in the first equation, yet it is used as a known function within the loss in the second equation.**
>
>
> Thank you for pointing this out. We have revised Equations (5) and (6) to avoid the circular definitions. The revised formulas are as follows:
>
> \begin{align}
> \Phi_{a} &= \underset{\eta}{\arg\min}\ L_{\eta}(F_{GT}, F_{a}),
> \end{align}
>
> \begin{align}
> L_{\eta} &= \lVert F_{GT} - \eta(F_{a}) \rVert_{2}.
> \end{align}
>
> **Q5: In Equation (8), is there a missing negative sign? In $L_{m,m'} = \sum_{B} \sum_{c} \sum_{B'} \log\left( \frac{\exp(s_{B,c,B'})}{\sum_{l} \exp(s_{B,c,B'})} \right)$, is the summation in the denominator performed over all $l$? However, the term inside the exponential function being summed is $\exp(s_{B,c,B'})$, which uses the index $B'$. Perhaps the correct formula is something like $L_{m,m'} = -\sum_{B} \sum_{c \in \text{cells}(B)} \log \frac{\exp\left(s_{B,c,B}\right)}{\sum_{l \in \mathcal{B}} \exp\left(s_{B,c,l}\right)}$.**
>
> Thank you very much for pointing out the error and helping us correct it. Our method involves a large number of symbols, which unfortunately introduced some errors and may have caused confusion during reading. The formula you provided is correct; we have changed it to:
>
> \begin{equation}
>     L_{m,m'} = -\textstyle \sum_{B \in \mathcal{B}} \sum_{c \in cells(B)} \mbox{log}\Big(\frac{\mbox{exp}(s_{B,c,B})}{\sum_{l \in \mathcal{B} } \mbox{exp} (s_{B,c,l})}\Big).
> \end{equation}
> where $\mathcal{B}$ is the set of objects.
>
> Besides, to avoid misunderstanding, we replace $B'$ in Equation 7 with $P$, which can represent the bounding box of any object, including $B$.

---

> ### Author Response · Authors · 2025-11-27
> **Response to Reviewer w8N3 [3/3]**
>
> **D1: The citation format in the paper is clearly inconsistent. For example, in Table 2, it is evident that some papers have year information while others do not. It is recommended that the authors re-check and unify the citation format.**
>
> Thank you for pointing out the error. We have re-checked the citation format and made the necessary corrections.
>
> **D2: It is recommended that the authors provide a detailed overview of the GT-Space framework in Section 3.2. This should include a comprehensive explanation of the workflows for pre-training, training, inference, and the process of integrating a new agent. The input, specific operations, output, and the meaning of each stage should be clearly articulated.**
>
> Thank you for your suggestion. Although we have provided implementation details in the appendix, we agree that a more detailed overview in Section 3.2 is necessary. In the revised version, we have expanded Section 3.2 with a more detailed explanation of our proposed framework, including the workflows for pre-training, training, inference, and the process of integrating a new agent.
>
> **D3: It is recommended that the authors provide more detailed explanations and descriptions for the formulas and symbols used in the paper, especially when new symbols are introduced. If the code is not open-sourced, the formulas in the paper must be sufficiently comprehensive to ensure that readers can understand the method.**
>
> In this paper, we employ many formulas and symbols, which may introduce some difficulty in reading; however, these components are necessary for clearly presenting our method. To further improve clarity, we provide a consolidated notation table that summarizes their definitions and roles, which is shown in Tab.5 in the Appendix.
>
> **D4: It is suggested to add a comparison of system complexity in the "Newly Added Agent" section.**
>
> Thank you for your suggestion. As with Question W3, we have added the comparison of system complexity in the "Newly Added Agent".
>
> **D5: It is suggested to include a sensitivity analysis for parameters such as the temperature $\tau$.**
>
>
> In contrastive learning, the temperature is used to adjust the “sharpness” of the similarity distribution, thereby controlling how strongly the model distinguishes between positive and negative samples. We evaluate the model’s performance under different temperature settings.
> Based on empirical practice in multimodal tasks, the temperature parameter is typically set within the range of 0.05–0.1. Therefore, we evaluate three settings: 0.05, 0.07, and 0.1. The results in the table below show that a temperature of 0.1 yields the best performance. This is because smaller temperatures make the distribution sharper and strengthen the separation between positive and negative samples, but lead to training instability. Hence, 0.1 serves as a more optimal choice.
>
>
> | **Dataset** | **Temperature** | **A1** | **A2** | **A3** | **A4** |
> |-------------|----------------|--------|--------|--------|--------|
> | **OPV2V**   | 0.05           | 0.845  | 0.835  | 0.830  | 0.826  |
> |             | 0.07           | 0.850  | 0.842  | 0.824  | 0.830  |
> |             | 0.1            | 0.858  | 0.846  | 0.828  | 0.833  |

---

### Official Review · Reviewer_AVxH · 2025-10-28

**Soundness:** 2
**Presentation:** 3
**Contribution:** 3
**Rating:** 6
**Confidence:** 4

**Summary:**

This paper proposes GT-Space, a novel framework that constructs a common feature space from ground truth labels, offering a unified reference for feature alignment among heterogeneous agents. This approach eliminates the need for pairwise interactions between agents, enhancing scalability. Extensive experiments demonstrate that GT-Space achieves state-of-the-art performance in detection accuracy.

**Strengths:**

1.Common Feature Space for Heterogeneous Agents: GT-Space introduces a common feature space for aligning heterogeneous agents, which promotes the practical deployment of heterogeneous collaborative perception systems.

2.Contrastive Learning for Consistent Representation: By employing contrastive learning to supervise the fusion network, GT-Space encourages different agents to learn consistent feature representations for the same instances, improving robustness.

3.Superior Performance Across Modality Combinations: Experimental results clearly indicate that GT-Space achieves advanced perception performance across various heterogeneous modality pairings, showcasing its effectiveness.

**Weaknesses:**

1.Generalization to Unseen Agent Types: The paper trains and tests with the same set of agent types (e.g., agents A1-A4 are used for both training and testing under various combinations). However, in real-world scenarios, new, unseen agent types may need to be integrated into the system. Without ground truth labels for these new agents, how can the projection layer be trained to adapt to the fusion model for collaborative perception? What would be the estimated amount of training data and training duration required for such adaptation?

2.Handling Misaligned Feature Spaces in Real-World Scenarios: GT-Space trains the model using observations from a single agent that assumes spatially aligned data. In practical settings, the feature spaces of different agents are likely to be misaligned. While coordinate transformations can achieve spatial alignment, the resulting features might differ significantly from those used during training (e.g., containing more background features). How does the proposed method address this discrepancy and maintain performance robustness in such real-world, misaligned scenarios?

**Questions:**

See the weaknesses.

---

> ### Author Response · Authors · 2025-11-27
> **Response to Reviewer AVxH**
>
> We thank you for your time and effort in reviewing our paper. Below, we respond to address your concerns.
>
> **W1: Generalization to Unseen Agent Types: The paper trains and tests with the same set of agent types (e.g., agents A1-A4 are used for both training and testing under various combinations). However, in real-world scenarios, new, unseen agent types may need to be integrated into the system. Without ground truth labels for these new agents, how can the projection layer be trained to adapt to the fusion model for collaborative perception? What would be the estimated amount of training data and training duration required for such adaptation?**
>
>
> For newly introduced, unseen agents, the fusion network remains fixed, and only their respective projection layers are trained.
>
> In our framework, introducing unseen agents does not require retraining the fusion network. For newly introduced agents, the fusion network remains fixed, and only their specific projectors are trained. This allows new agents to adapt to the existing fusion model and participate in collaboration. During this process, no additional ground-truth labels are required for training the projectors of the new agents, as the data used to train the fusion network previously can be reused.
>
> To ensure effectiveness, we train the projectors using the complete OPV2V dataset in our experiments. However, since the projectors are lightweight, their training complexity is relatively low. In the updated version, we provide the training costs in the Fig. 9 in the Appendix, and the results demonstrate the efficiency of our approach.
>
>
>
>
> **W2: Handling Misaligned Feature Spaces in Real-World Scenarios: GT-Space trains the model using observations from a single agent that assumes spatially aligned data. In practical settings, the feature spaces of different agents are likely to be misaligned. While coordinate transformations can achieve spatial alignment, the resulting features might differ significantly from those used during training (e.g., containing more background features). How does the proposed method address this discrepancy and maintain performance robustness in such real-world, misaligned scenarios?**
>
>
> During training, the employed dataset was meticulously annotated. Specifically, prior spatial information from LiDAR point clouds was utilized to generate 3D bounding boxes, which were subsequently subjected to rigorous manual verification to ensure the accuracy and reliability of the annotations [1].
>
> For the real-world deployments, spatial misalignment is indeed unavoidable.
> Thus we introduce a robustness experiment to evaluate the model's performance under localization errors. Specifically, we add Gaussian noise into the accurate poses to simulate spatial errors, where the pose noise is set to $N(0, \sigma_p^2)$ on x,y location and $N(0,\sigma_r^2)$ on yaw angle [1].
> The experimental results are shown in Fig. 5 (a-b) of the updated version.
> The results demonstrate that our method consistently maintains state-of-the-art performance across a wide range of pose error conditions.
>
>
> [1] Ruiyang Hao, et al. Rcooper: A real-world large-scale dataset for roadside cooperative perception. Proceedings of the IEEE/CVF conference on computer vision and pattern recognition. 2024.

---

### Official Review · Reviewer_BvnQ · 2025-11-01

**Soundness:** 3
**Presentation:** 3
**Contribution:** 3
**Rating:** 6
**Confidence:** 5

**Summary:**

GT-Space is a heterogeneous collaborative perception framework that avoids pairwise feature alignment between agents with different sensors/encoders. It builds a ground-truth–derived common BEV feature space from 3D box labels and learns a single lightweight projector per agent to map local features into this space. On OPV2V, V2XSet, and a real-world RCooper scene, GT-Space consistently outperforms end-to-end, interpreter, and retraining baselines, especially boosting weaker/camera agents. Ablations show the projector and GT-space supervision are key contributors to gains.

**Strengths:**

1. GT-space requires only one projector per agent, without pairwise adapters or encoder retraining which is scalable.

2. GT-space uses contrastive learning to handle the bottleneck effect in collaborative perception.

3. The experiments show that GT-space has SOTA performance among widely used datasets.

**Weaknesses:**

This work requires accurate, dense 3D labels to build the common space, which is often impractical in real-world deployments.

The reviewer suggests adding robustness experiments, such as how the pipeline handles communication latency and localization errors.

This work evaluates only 3D detection; performance on other tasks, such as lane segmentation or tracking, is unclear.

**Questions:**

What is the advantage of this work over interpreter-based methods? The author claimed that they are not scalable but did not explain why.

---

> ### Author Response · Authors · 2025-11-27
> **Response to Reviewer BvnQ [1/2]**
>
> Thank you for your constructive comments and positive feedback on our submission. We address your concerns as follows:
>
> **W1: This work requires accurate, dense 3D labels to build the common space, which is often impractical in real-world deployments.**
>
> Accurate 3D labels are critical not only for constructing a common feature space, but also for any perception method. For example, SAM 3D [1] proposes a generative model for visually grounded 3D object reconstruction, predicting geometry, texture, and layout from a single image.
> Since it is difficult for humans to directly annotate 3D shapes, this model can generate multiple 3D candidate results, from which human annotators select the best-matching one. VGGT [2] proposes a large transformer that directly infers all key 3D attributes of a scene, including camera parameters, point maps, depth maps, and 3D point tracks, from one, a few, or hundreds of its views. Therefore, obtaining accurate 3D annotations in real-world scenarios is feasible.
>
> Besides, as reported in the appendix, we evaluate our approach on the RCooper real-world dataset. The results summarized in Tab. 7 demonstrate the effectiveness of our method in real-world scenarios.
>
> [1] Xitong Yang et al. SAM 3D Body: Robust Full-Body Human Mesh Recovery, arXiv preprint, 2025.
>
> [2] Jianyuan Wang, et al. Vggt: Visual geometry grounded transformer. Proceedings of the Computer Vision and Pattern Recognition Conference. 2025.
>
>
> **W2: The reviewer suggests adding robustness experiments, such as how the pipeline handles communication latency and localization errors.**
>
>
> Thank you for your suggestion. We agree that conducting experiments under communication delay and localization error is necessary. We introduce a robustness experiment against communication latency and localization errors.
>
> For localization error, we add Gaussian noise into the accurate poses, where the pose noise is set to $N(0, \sigma_p^2)$ on x,y location and $N(0,\sigma_r^2)$ on yaw angle [3].
> The experimental results are shown in Fig. 5 (a-b) of the updated version.
> The results demonstrate that our method consistently maintains state-of-the-art performance across a wide range of pose error conditions.
>
> For communication latency, we configure different latency values ranging from 100 ms to 500 ms [4] and the results are shown in Fig. 5 (c-d). It can be seen that even with a 500~ms latency, our method still outperforms the baseline methods.
>
> [3] Yifan Lu, et al. Robust collaborative 3d object detection in presence of pose errors. IEEE International Conference on Robotics and Automation (ICRA) 2023.
>
> [4] Haibao Yu, et al. Flow-based feature fusion for vehicle-infrastructure cooperative 3d object detection. Advances in Neural Information Processing Systems 2023.
>
> **W3: This work evaluates only 3D detection; performance on other tasks, such as lane segmentation or tracking, is unclear.**
>
> Our evaluation is conducted on the 3D detection task. This is because the common feature space we construct relies on 3D ground-truth information. To extend the framework to other tasks, one would need to use the corresponding ground-truth information to construct the appropriate feature space.
> For the lane segmentation task, lane labels need to be incorporated during the training of the ground-truth encoder, and a specific segmentation decoder is employed to output the results.
>
> For the tracking task, we consider the object bounding box information to be sufficient. Therefore, we conduct experiments on the RCooper [5] dataset, which adopts AB3Dmot [6] tracker to achieve efficient and high-quality tracking based on detection results.
>
> We adopt the evaluation metrics in [7], 1) average multi-object tracking accuracy (AMOTA) and 2) average multi-object tracking precision (AMOTP). As shown in the table below, our method outperforms all baseline models, which is consistent with the 3D detection results.
>
> | Method     | AMOTA | AMOTP |
> |------------|-------|-------|
> | No fusion  | 0.083 | 0.227 |
> | HM-ViT     | 0.224 | 0.354 |
> | PnPDA      | 0.228 | 0.357 |
> | Hetecooper | 0.227 | 0.361 |
> | HEAL       | 0.232 | 0.364 |
> | STAMP      | 0.231 | 0.359 |
> | GT-Space   | 0.236 | 0.367 |
>
> [5] Ruiyang Hao, et al. Rcooper: A real-world large-scale dataset for roadside cooperative perception. Proceedings of the IEEE/CVF conference on computer vision and pattern recognition. 2024.
>
> [6] Xinshuo Weng, et al. Ab3dmot: A baseline for 3d multi-object tracking and new evaluation metrics. arXiv preprint arXiv:2008.08063 (2020).
>
> [7] Xinshuo Weng, et al. 3d multi-object tracking: A baseline and new evaluation metrics. 2020 IEEE/RSJ International Conference on Intelligent Robots and Systems, 2020.

---

> > ### Author Response · Authors · 2025-11-27
> > **Response to Reviewer BvnQ [2/2]**
> >
> > **Q1: What is the advantage of this work over interpreter-based methods? The author claimed that they are not scalable but did not explain why.**
> >
> > Our work has several advantages over interpreter-based methods:
> >
> > 1. A centralized fusion network, which mitigates the bias arising from multiple feature transformations performed by interpreters.
> >
> > 2. The integration of ground-truth information for explicit supervision, enabling direct alignment of object-relevant features and thereby enhancing the effectiveness of feature alignment.
> >
> > 3. Ground-truth data also serves to enhance feature representations, improving robustness to under-performing agents.
> >
> > Interpreter-based methods essentially map the features of other modalities into the ego agent’s feature space, i.e., aligning them to the ego’s semantics. This method requires each collaborative agent to train a pair-specific interpreter for the ego agent, making it difficult to scale in deployment.

---

### Official Review · Reviewer_dxid · 2025-11-03

**Soundness:** 4
**Presentation:** 3
**Contribution:** 3
**Rating:** 8
**Confidence:** 5

**Summary:**

This paper introduces GT-Space, which is a scalable collaborative perception framework for heterogeneous autonomous driving agents. GT-Space uses a common feature space constructed from ground-truth labels to serve as a unified reference for feature alignment and a lightweight projector for each agent to map its features into that space. It achieves SOTA performance on OPV2V, V2XSet, and RCooper dataset.

**Strengths:**

1. The idea of aligning all heterogeneous features in the GT space to construct a common, unified feature space is well-motivated and novel.
2. The designed system is highly scalable and flexible. Because individual agents' encoders and detection heads are kept frozen, a new or unseen agent can be integrated simply by training a single, lightweight projector module to map its features to the GT-Space.
3. The framework's performance is not bottlenecked by the capability of the ego agent. By using the GT-Space as a strong, objective reference, the fusion network can effectively leverage complementary strengths and compensate for weaker agents (like camera-only models).
4. The paper demonstrates SOTA performance on multiple benchmarks.

**Weaknesses:**

1. I have a major concern in the scalability of fusion training. The fusion network is trained using a "combinatorial contrastive loss" across all pairs of modalities. The paper gives an example with 3 models, resulting in 3 pairs. This implies that for $M$ distinct modality types, the training complexity is $O(M^2)$. This is not scalable
2. The visualization in Figure 5 is not very insightful. It simply shows that the fused feature map has stronger activations than the original. More compelling visualizations would have been:A visualization of the GT-Space $F_{GT}$ itself.A side-by-side comparison of a projected camera feature $\Phi_{cam}(F_{cam})$ and a projected LiDAR feature $\Phi_{lidar}(F_{lidar})$ to visually demonstrate the alignment quality in the common space.

**Questions:**

I would like to raise my rating if the authors can address my concerns shown in weakness section.

---

> ### Author Response · Authors · 2025-11-27
> **Response to Reviewer dxid**
>
> We sincerely appreciate the reviewer's constructive feedback and positive remarks on our work. We provide the following detailed responses to your major concerns.
>
> **W1: I have a major concern in the scalability of fusion training. The fusion network is trained using a "combinatorial contrastive loss" across all pairs of modalities. The paper gives an example with 3 models, resulting in 3 pairs. This implies that for $M$ distinct modality types, the training complexity is $O(M^2)$. This is not scalable.**
>
> Your concern is reasonable. During training, taking all possible pairs into account would result in very high complexity as the number of modalities grows. Therefore, the extension of our model is not achieved by retraining the fusion network.
> For newly added agents, we freeze the fusion network and only train their specific projectors, enabling them to adapt to the fusion network and participate in collaboration. Hence, the training complexity is $O(M)$. The results are shown in the Tab. 9 in the Appendix.
>
>
> **W2: The visualization in Figure 5 is not very insightful. It simply shows that the fused feature map has stronger activations than the original. More compelling visualizations would have been:A visualization of the GT-Space $F_{GT}$ itself.A side-by-side comparison of a projected camera feature $\Phi_{cam}(F_{cam})$ and a projected LiDAR feature $\Phi_{lidar}(F_{lidar})$ to visually demonstrate the alignment quality in the common space.**
>
> Thanks for your suggestions. We have added the visualization of ground truth feature, projected camera feature, and projected LiDAR feature in Fig.7 in the updated version. Besides, we provide more visualization examples of the detection results in the appendix of the updated version.

---

### Author Response · Authors · 2025-12-02
**A Summary of Our Contributions and Responses**

Dear Reviewers and ACs,

We sincerely thank you for the time and effort you have dedicated to reviewing our work, as well as for the constructive questions and comments you provided. These insights have greatly contributed to improving the quality of our paper. We are pleased to see that our contributions have been appreciated by all reviewers.

In the following, we would like to summarize the contributions and revisions of this paper again.

**Contributions:**

- **[High-level insight]** We enhance heterogeneous collaborative perception by introducing a ground-truth feature space that serves as a unified common space for feature alignment. This perspective is `well-motivated` (dxid), `clear` (w8N3), `novel` (dxid and AVxH), and `promotes the practical deployment` (AVxH).

- **[Flexible and scalable framework]** Our proposed GT-Space constructs a common feature space using ground-truth labels and aligns heterogeneous features to this space for fusion. Each agent performs alignment through its projector, and contrastive learning is employed to ensure consistent feature representations across agents for the same instances, enhancing robustness. This design is `flexible` (dxid) `scalable`  (dxid, BvnQ) `robust` (AVxH), and `plug-and-play` (w8N3).

- **[Comprehensive experiments]**  GT-Space is supported by extensive and `clear` (AVxH) experimental evidence on `multiple` (dxid) and `widely used` (BvnQ) datasets, covering both simulated and real-world scenarios. Ablation studies further confirm that projector and GT-space supervision are `key contributors to gains` (BvnQ).

**Responses and Revisions**

- **[For Reviewer dxid]**
  - We clarify that the mechanism for newly added agents does not require high complexity.
  - The updated paper now presents more insightful visualizations to better convey the results.

- **[For Reviewer BvnQ]**
   - We discuss the feasibility of obtaining accurate perception data at low cost.
   - We add experiments to evaluate the robustness under communication latency and localization errors.
   - We extend our evaluation to the tracking task, in addition to 3D detection.
   - We clarify the advantages of our method over the Interpreter-based method.

- **[For Reviewer AVxH]**
   - We explain the mechanism for newly added agents and report the training cost in the updated paper.
   - We introduce a robustness experiment to evaluate the model's performance under localization errors.

- **[For Reviewer w8N3]**
   - We conduct experiments to verify the robustness of our method under distribution shifts in objects.
   - We explicitly report the comparison of training computation and time for adding new agents.
   - We adjust the contrastive loss function to ensure balanced weighting across objects of varying sizes. The effectiveness is validated through experiments.
   - Our method inevitably involves numerous symbols, which leads to some mistakes. With the reviewer’s valuable comments, we have corrected these mistakes and added a table summarizing the symbols, thereby clarifying the notation system.
   - We modify some potentially ambiguous figures to improve clarity.
   - We add a detailed explanation of the workflows of our proposed framework to improve readability.
   - We add a sensitivity analysis for the temperature in the contrastive learning.


All modifications have been highlighted in blue in our revised manuscript. Thanks again for your efforts in reviewing our work, and we hope our responses can address any concerns about this work.

Thanks

The Authors of Submission 10421.

---

### Meta-Review · Area_Chair_mwqi · 2026-01-04

**Summary:**

Across the four reviews, the overall assessment is favorable regarding motivation and empirical performance. All reviewers describe GT-Space as a heterogeneous collaborative perception framework that constructs a ground-truth–derived common BEV feature space and learns a lightweight projector per agent to map local features into this space for fusion and detection. Reviewers highlight that freezing agent-specific encoders/heads and training only per-agent projectors makes the framework scalable and flexible, and the paper reports strong results on widely used benchmarks (OPV2V, V2XSet, and RCooper), with particular benefits for weaker agents. The main concerns informing the decision centered on (i) scalability of the contrastive training formulation as the number of modality types grows, (ii) practical dependence on accurate/dense 3D labels and robustness to real-world issues (distribution shift, localization error, communication latency), (iii) generalization to newly added/unseen agents and the deployment/training cost of adaptation, and (iv) clarity/reproducibility issues (notation, figure inconsistencies, and implementation details). The rebuttal provides clarifications and adds several targeted experiments and revisions (new visualizations, robustness tests, cross-dataset evaluation, tracking evaluation, training-cost reporting for adding agents, and fixes to notation/formulas/figures), addressing many of the decision-critical points, though label reliance and the scope of evaluated tasks/settings remain as inherent constraints.

**Reviewer Concerns:**

Concerns addressed by the rebuttal:
1. Scalability of contrastive training across modality pairs (dxid): The authors clarify that extension to newly added agents does not require retraining the fusion network; the fusion network is frozen and only the new agent’s projector is trained, reducing the training complexity for adding agents. They report results for this setting (Tab. 9 in the Appendix).
2. More informative visualizations (dxid): The updated paper adds visualizations of the ground-truth feature, projected camera feature, and projected LiDAR feature (Fig. 7), plus additional detection visualizations in the appendix.
3. Robustness to localization errors and communication latency (BvnQ, AVxH): The rebuttal adds experiments with pose noise (Gaussian noise on x,y and yaw) and latency ranging from 100ms to 500ms, reporting that GT-Space maintains strong performance across these conditions (Fig. 5).
4. Distribution shift / cross-dataset robustness (w8N3): The authors report evaluation where the fusion model trained on OPV2V is tested on V2XSet, and provide comparative results showing GT-Space remains competitive under this shift.
5. Beyond 3D detection: tracking evaluation (BvnQ): The rebuttal extends evaluation to a tracking task on RCooper using AB3Dmot and reports AMOTA/AMOTP comparisons against multiple baselines, with GT-Space achieving the highest reported values in the provided table.
6. Advantages over interpreter-based methods (BvnQ): The authors clarify their claimed advantages, including use of a centralized fusion network, explicit supervision via ground-truth information for alignment, robustness benefits for under-performing agents, and improved scalability versus pair-specific interpreters that map into an ego feature space.
7. Generalization to newly added agents and adaptation cost (AVxH, dxid, w8N3): The rebuttal clarifies the mechanism (freeze fusion network; train only the new projector) and explicitly reports training computation/time for adding new agents (Fig. 9 in Appendix), positioning the projector training as low cost due to its lightweight design.
8. Clarity/notation/figure and formula issues (w8N3): The authors correct figure inconsistencies (Figure 1 detection head depiction; Figure 2 loss placement), revise vague/circular equations (5–6), fix an error in Equation (8), unify citation formatting, add a consolidated notation table (Tab. 5), and expand Section 3.2 to describe workflows (pre-training, training, inference, and adding a new agent). They also commit to open-sourcing code.
9. Loss imbalance across object sizes (w8N3): The authors adjust the contrastive loss to balance weighting across objects of varying sizes and show improved results on RCooper (table comparing “GT-Space” vs “GT-Space (average)”).
10. Sensitivity to temperature in contrastive learning (w8N3): The rebuttal adds a temperature sensitivity analysis (0.05/0.07/0.1) and reports best performance at 0.1 on OPV2V.

Concerns partially addressed or still outstanding:
1. Dependence on accurate/dense 3D labels (BvnQ, w8N3): The rebuttal argues that accurate 3D annotations are feasible (citing recent works) and notes evaluation on a real-world dataset (RCooper). However, the method still fundamentally relies on ground-truth labels to construct the common space during training, and the practicality of obtaining such labels at scale remains an inherent requirement of the approach as presented.
2. Upper bound vs. end-to-end fine-tuning (w8N3): The authors justify freezing encoders/heads as a trade-off to preserve standalone perception and reduce training cost, but this does not remove the limitation that end-to-end fine-tuning could potentially yield higher collaborative performance under certain assumptions.
3. Task/setting scope beyond reported experiments: While the rebuttal adds tracking and multiple robustness tests, the evaluation remains centered on BEV-based 3D perception setups and the datasets/tasks explicitly reported.

**Reviewer Scores:**

Reviewer dxid: Likely unchanged. The reviewer is already strongly positive and explicitly indicated willingness to raise their rating if concerns are addressed; the rebuttal responds directly on scalability for new agents and improves visualizations.

Reviewer BvnQ: Likely to increase slightly. The rebuttal adds the requested robustness experiments (latency/localization), extends to tracking with reported gains, and clarifies the comparison to interpreter-based methods; label-dependence remains but is at least discussed and partially mitigated by real-world evaluation.

Reviewer AVxH: Likely unchanged or slightly higher. The rebuttal clarifies the “new agent” integration mechanism (freeze fusion; train projector), reports training cost, and adds localization-error robustness experiments addressing misalignment concerns.

Reviewer w8N3: Likely to increase modestly. Many of this reviewer’s concerns were about missing robustness quantification, missing “add new agent” training-cost reporting, unclear implementation/notation, figure/formula errors, and loss re-weighting; the rebuttal introduces cross-dataset results, adds cost comparisons, corrects multiple concrete errors, provides a notation table and workflow explanation, adds temperature sensitivity analysis, and validates the revised loss weighting empirically.

---

### Decision · Program_Chairs · 2026-01-26

Accept (Poster)